

# Snow observations in Mount-Lebanon (2011-2016)

Abbas Fayad[1,2], Simon Gascoin[1], Ghaleb Faour[2], Pascal Fanise[1], Laurent Drapeau[1], Janine Somma[3], Ali Fadel[2], Ahmad Al Bitar[1], and Richard Escadafal[1]

[1]Centre d'Etudes Spatiales de la Biosphère (CESBIO), UPS/CNRS/IRD/CNES, Toulouse, France
[2]National Council for Scientific Research/Remote Sensing Center (CNRS/NCRS), Beirut, Lebanon
[3]Saint Jospeh University, Beirut, Lebanon

*Correspondence to*: Abbas Fayad (abbas.fayad@gmail.com)

**Abstract.** We present a unique meteorological and snow observational dataset in Mount-Lebanon, a mountainous region with a Mediterranean climate, where snowmelt is an essential water resource. The study region covers the recharge area of 
10 three karstic river basins (total area of 1092 km$^2$ and an elevation up to 3088 m). The dataset consists of: (1) continuous meteorological and snow height observations; (2) snowpack field measurements; and (3) medium resolution satellite snow cover data. The continuous meteorological measurements at three automatic weather stations (MZA 2296 m, LAQ 1840 m, and CED 2834 m a.s.l.) include surface air temperature and humidity, precipitation, wind speed and direction, incoming and reflected shortwave irradiance, and snow height, at 30 minute intervals for the snow seasons (November – June) between 
15 2011 and 2016 for MZA and 2014–2016 for CED and LAQ. Precipitation data was filtered and corrected for Geonor undercatch. Observations of snow height (HS), snow water equivalent, and snow density were collected at 30 snow courses located at elevations between 1300 and 2900 m a.s.l. during the two snow seasons 2014–2016 with an average revisit time of 11 days. Daily gap-free snow cover extent (SCA) and snow cover duration (SCD) maps derived from MODIS snow products are provided for the same period (2011–2016). We used the dataset to characterize mean snow height, snow water equivalent 
(SWE), and density for the first time in Mount-Lebanon. Snow seasonal variability was characterized with high HS and SWE variance and a relatively high snow density mean equal to 467 kg m$^{-3}$. We find that the relationship between snow depth and snow density is specific to the Mediterranean climate. The current model explained 34% of the variability in the entire dataset (all regions between 1300 and 2900 m a.s.l.) and 62% for high mountain regions (elevation 2200–2900 m a.s.l.). The dataset is suitable for the investigation of snow dynamics and for the forcing and validation of energy balance
models. Therefore, this data set bears the potential to greatly improve the quantification of snowmelt and mountain hydrometeorological processes in this data-scarce region of the Eastern Mediterranean. The doi for the data is: doi.org/10.5281/zenodo.321405.

## 1 Introduction

Water scarcity is a growing concern in Lebanon due to the unsustainable water resource management, the limited 
accessibility to the water sources, the increased water demand by all sectors, the increased water pollution, and sea water



intrusion (MOEW, 2010; UNDP, 2014; Kalaoun et al. 2015). Lebanon receives on average 830 mm of precipitation per water year (September–August) (MOEW, 2010). Most precipitation falls during winter season (December–March). The two Lebanese mountain chains, the Mount- and Anti-Lebanon, receive between 50 to 67% of the total annual precipitation as snow (UNDP, 2014). This is mainly due to the orographic enhancement of the precipitation on the western slope of the Mountain chains. The Mount-Lebanon range has an average elevation above 2200 m and stretches over a distance of 150 km parallel to the Mediterranean coast, therefore causing enhanced orographic uplift of moist air masses. Due to the influence of the Mediterranean climate (wet winter, dry summer) most of the precipitation above 1200 m a.s.l. falls as snow on Mount-Lebanon (Shaban et al., 2004; Aouad-Rizk et al., 2005; Mhawej et al., 2014; UNDP, 2014). The average contribution of snowmelt to spring and river discharge in Mount-Lebanon was estimated at 30% by Telesca et al. (2015). Snowmelt contributes to the recharge of karstic aquifers and springs of all the watersheds located in the windward regions of Mount-Lebanon (Fig. 1) (e.g., Bakalowicz et al., 2007; Doummar et al., 2014). This snow contribution to groundwater recharge can reach up to 75% in the upper mountainous aquifers where the groundwater recharge was estimated at 81% from precipitation in the snow dominated regions of the El Kelb Basin (Margane et al., 2013; Königer and Margane, 2014). The snowmelt contribution from high elevation regions (above 1800 m a.s.l.) was estimated to contribute to around 56% of the major spring discharge at the lowland regions (Margane et al., 2013). The coastal watersheds, such as the EL Kelb Basin, are the major sources of water for the coastal population, where most of the Lebanese population is located.

Although snow is recognized as a major component of the hydrologic system in Mount-Lebanon (Shaban et al., 2004; Aouad-Rizk et al., 2005; Bakalowicz et al. 2008; Mhawej et al., 2014; Koeniger and Margane 2014), the link between snowmelt and the hydrological processes remains poorly characterized at the basin scale. This can be attributed to the (1) lack of operational snow observation network in Lebanon and (2) limited number of published basin scale hydrometeorological datasets. Meteorological stations operated by the department of civil aviation are usually located below snowline (maximum elevation at 1220 m a.s.l.) and the few stations located in the mountainous regions (elevation range 1510–1890 m a.s.l.) are not equipped to measure snowfall. Furthermore, only a few number of datasets on precipitation, temperature, snow, groundwater recharge and streamflow were made available for basin scale studies (e.g., Koeniger and Margane 2014). Existing datasets, available through published material, are usually limited to: (1) national scale studies with monthly or yearly means (Shaban et al. 2004; Corbane et al., 2005; Mhawej et al., 2014; Telesca et al., 2014); (2) multi-years daily spring discharge and precipitation time series (e.g., Hreiche et al., 2007); (3) seasonal observations collected for meso-scale catchment studies (Bernier et al. 2003; Aouad-Rizk et al., 2005; Doummar et al. 2014); and (4) local scale snowpack observations (e.g., Somma et al., 2006, 2014). In most cases the research datasets are not made publicly available.

In this paper, we present a dataset targeted at the study of the mountain and snow hydrology in three meso-scale basins (area 256–513 km$^2$) located on the west slope of Mount-Lebanon (Fig. 1). The dataset consists of: (1) continuous meteorological and snow height observations collected at three AWS (2011–2016); (2) snowpack field measurements collected during two snow seasons (2014–2016); and (3) medium resolution satellite observations of the snow cover extent at a daily time step (MODIS). We use these data to characterize the variability of key snowpack properties. The dataset



presented in this paper is unique because it is the only dataset that includes a range of continuous snow and meteorological measurements in the mountain region of Lebanon at the elevation regions between 1300 and 2900 m a.s.l.. The data presented are readily suitable for the forcing and validation of a snowpack energy and mass balance model. The data also can be useful for further hydro-meteorological studies such as climate model downscaling or hydrological modeling for water

5 resource management.

The study area is described in section 2. Meteorological observations and post processing methods, snow course measurement protocols, and MODIS data processing are presented in section 3. Section 4 provides a summary of the observations and an example application on using the datasets to derive the relationships between HS, snow water equivalent (SWE) and snow density. Data accessibility and conclusions are presented in sections 5 and 6 respectively.

## 10 2 Study area

This study measurements were collected in the upper area of three meso-scale snow dominated mountain basins located in Mount-Lebanon with an average centroid located at 34.10 N and 35.90 E covers a total area of 1092 km$^2$ (Fig. 1). These basins belong to the "coastal watersheds", which supply fresh water to major Lebanese cities including Beirut. Due to the influence of the Mediterranean climate, most precipitation falls between November and April. Winter precipitation

15 (December–March) accounts for 84 % of the total annual precipitation. Most precipitation above 1600 m a.s.l. falls as snow. The topography of the mid-elevation regions (1600–2200 m a.s.l.) are usually rugged terrain (Fig. 2). The mid-elevation and high elevation plateau are found at elevation ranges between 2300–2500 m and 2700–3000 respectively (Fig. 2). The treeline is located at 1550 m a.s.l. where sparse scrublands dominate most of the land cover. Most snow-fed karstic springs are located at altitudes between 300 and 2280 m a.s.l. The physical attributes of the three study basins are shown in Table 1.

## 20 3 Data description and methods

### 3.1 Meteorological data

The three automatic weather stations (AWS) were installed in Mount–Lebanon above the winter snowline (approximately 1550 m a.s.l.) with the primary objective to monitor the meteorological variables that drive seasonal snowpack evolution on Mount–Lebanon (Fig. 1 and Fig. 2). The Laqlouq AWS (LAQ) is located in a monastery at 1840 m a.s.l., the Mzar station

25 (MZA) is located in a ski resort domain at 2296 m a.s.l., while the Cedars AWS (CED) (2834 m a.s.l.) is located on the higher plateau below the mountain's peak at Qornet El Sawda, 3088 m a.s.l. (Fig. 1, Table 2). The LAQ station is in a relatively flat plain area with fruit trees, bare rocks, and sparse short–grasslands. The MZA station is located on one of the medium elevation peaks in a rugged terrain mountainous region (maximum elevation in the area is ~2600 m a.s.l.) with dominant bare soils and sparse speargrass grassland. The CED AWS is located on a higher plateau with dominant bare rocks

30 and sparse shrubs and grasslands at lower elevations (1600–2200 m a.s.l.). Wind effects on snow cover are more noticeable



in MZA due to the combined effect of topography and higher wind velocities. The three stations are operated under a joint–program for establishing a network for snow observation (NSO). The program, established in 2010, is a collaboration between the 'Institut de Recherche pour le Développement' IRD (France), the 'Centre d'Etudes Spatiales de la Biosphere' CESBIO (France), the National Council for Scientific Research – Remote Sensing Center (CNRS\NCRS) (Lebanon), and the
University of Saint Joseph USJ (Lebanon).

Meteorological data are available since snow season (December–June) 2011 for Mzar AWS and the monitoring network became fully operational in snow season 2014–2015 with the installation of the third AWS (LAQ) at Laqlouq (Table 2). Meteorological data, including snow depth, temperature, relative humidity, incoming and reflected solar radiation, wind speed and direction, and atmospheric pressure, are collected at the three sites using sensors mounted to towers (Fig. 3). Each
station consists of a datalogger (CR1000; Campbell Scientific Inc., Utah, USA) and a precipitation gauge (T–200B; Geonor Inc., Eiksmarka, Norway), a snow depth sensor (SR50A; Campbell Scientific Inc., Utah, USA), an air temperature and humidity sensor (CS215; Campbell Scientific Inc., Utah, USA), an incoming and reflected solar radiation sensor (SP LITE 2 Pyranometer; Kipp & Zonen, Netherlands), a wind speed and direction sensor (Alpine v05103–45L; Young, USA). Data are transmitted via a GPRS modem every 8 hours. Observations from the three automatic weather stations (AWS) are collected
at 30 sec and then aggregated into 30 min averages. Temperature and humidity sensor were installed at 2.4, 3.9, and 4.2 m above ground level representing (MZA, LAQ, and CED). Wind speed sensors are installed at 2.6 m in MZA, 4.2 m in LAQ, and 4.9 m in CED. Snow depth observations were recorded automatically at each station using an acoustic snow gauge installed at 2.0 m in MZA and at 4.0 m in CED and LAQ. Precipitation data are recorded at a snow gauge placed in the proximity of the station (Fig. 3). Precipitation is being observed since 2012 at MZA (2012–2016) and since 2014 at LAQ
(2014–2016). The CED station is being equipped with a Geonor starting December 2016.  Data for precipitation were missing during the first year (2011–2012) at MZA and solar radiation measurements began in snow season 2014. CED AWS data between 24 April and 30 June 2015 were removed due to station rotation. The wind speed data located at CED were discarded after 15 January 2016 due to sensor malfunction. Missing data were less than 10% for all stations since the network became fully operational (2014–2016).

Raw data collected at the stations underwent basic quality control, including checks for missing data and boundary values. We performed further quality control for the 30 min and daily average observations by screening outliers and erroneous data following rules given by Serreze et al., (1999), Shafer et al., (2000), and Estévez et al., (2011).  Humidity, pyranometers, and wind sensors were unheated and thus may be subject to error when covered by frost (Malek, 2008). The accumulation of frost on sensor was observed during multiple field visits at MZA and CED. These events usually coincide
with the week following storm events.

For precipitation (P), air temperature (T), snow depth (SD), and humidity (H), we used running step tests to detect abrupt jumps in means, especially during storm events (Table 3). Incoming and reflected solar radiation measurements (SR) were screened using snow half–hourly albedo, by eliminating data that do not give a positive albedo ($0 \leq$ Albedo $\leq 1$). We used a positive snow height (HS) to detect the presence or absence of snow. Data ranges were used for P ($0 – 240$ mm h$^{-1}$), T (-30 –



+40 °C), H (0 – 100%), SD (0 – 4.5 m), SR (0 – 1500 W m$^{-2}$). For the two snow seasons 2015–2016 (November–June) data record retained at the three stations were 71.1, 95.4, and 94.3% of the total datasets for CED, LAQ, and MZA respectively. An example of the semi-hourly precipitation and temperature observations at Laqlouq for the snow seasons (2014-2016) are shown in Fig. 4.

## 3.2 Correcting for Geonor undercatch

The output data from the Geonor accumulating gauge was post-processed: to identify and correct biased observations, to determine the precipitation type, and to correct the Geonor precipitation undercatch. Three types of biases were found in the Geonor observations similarly to previous studies (Harpold and Pomeroy 2003; Pan et al. 2016): (1) erroneous readings associated with the Geonor field servicing (i.e. emptying and/or adding of antifreeze and oil to the Geonor bucket). (2) Jitters and diurnal noise due to wind speed (e.g., MZA) and changes in temperature are similar to those found in sites with strong diurnal changes in temperature, radiation, and wind speed (e.g. Harpold and Pomeroy 2003; Pan et al. 2016), and (3) long-term drift results from evaporation within the bucket, which occurs at the end of snow season when air temperature is high. We post-processed the raw precipitation data using a supervised correction similar to the one described in Harpold and Pomeroy, (2003) by performing the following steps: (1) the Geonor raw data was adjusted to account for when the gauge is emptied and/or filled with oil and antifreeze. (2) We used the predefined values for range test and cross validation tests (Table 3) to automatically flag and remove erroneous peaks (e.g. Psh > 120 mm). (3) All erroneous changes in the calculated raw cumulated precipitation (raw Psh < –20 mm and > 20 mm) was removed. (4) We flagged and removed all cumulated precipitation greater than 1000 mm (equal to the maximum capacity of the Geonor bucket), the precipitation data for the same time period was set to no-data in the final dataset. (5) All missing cumulative precipitation observations were assumed to be equal to the previous observed observation for running the filter. (6) We used a supervised rolling maximum filter (Harpold and Pomeroy, 2003) to remove the biased precipitation observation. The filter was run sequentially on the timeseries of the cumulated precipitation: the precipitation observation was retained if it is greater than the previous maximum.

The rolling maximum filter preserves the cumulative change in precipitation and the timing of precipitation events (Harpold and Pomeroy, 2003) (Fig. 5). However, a visual check is needed to flag potential errors. We visually compared the auto-filtered data versus the raw data to: (1) check for erroneous departures between the auto-filtered and raw data; (2) check if the filter captured the start of a precipitation and whether the calculated Psh occurred when humidity was greater than 80% (Psh with low humidity values were removed); and (3) check and correct errors attributed to gauge drift events, which are associated to evaporation effects, are not captured by the filter (Fig. 5a). We corrected these errors by manually replacing the filtered cumulated precipitation data to fit the actual change from the raw precipitation data (Fig. 5a). We made sure that the total sum of the replaced precipitation is equal to the cumulative observed precipitation which is assumed to be correct over the same time period. We also checked that the precise start and timing of precipitation events are preserved and that the long-term drift due to evaporation was eliminated (Fig. 5a).



The collection efficiency of precipitation gauges is influenced by the wind speed and a bias adjustment for solid precipitation is needed under windy conditions (Rasmussen et al., 2012; Buisan et al., 2016; Smith et al., 2016; Pan et al., 2016). Measurement errors due to gauge undercatch frequently range between 20% and 50% (Rasmussen et al., 2012). The catch efficiency for the Geonor with a single Alter-shield decreases linearly to approximately 60% at wind speed 6 m s$^{-1}$

(Rasmussen et al., 2001). The bias adjustment for precipitation undercatch is achieved by estimating the catch efficiency (CE) (wind–speed relationship) of the precipitation gauge. The determination of CE requires the determination of precipitation type and that the wind speed is measured at gauge height (Rasmussen et al., 2012). The need for precipitation type separation is important because the influence of wind is much more pronounced for solid precipitation than for liquid precipitation (Rasmussen et al., 2012). In this study, we applied bias correction for the filtered precipitation data ($P_{adj}$). For

solid precipitation, we used the empirical relationship between catch efficiency and wind speed derived by Thériault et al., (2012) after Yang et al., (1998) and Rasmussen et al., (2001):

$$P_{cor} = P_{adj}/CE \qquad (1)$$

$$CE = \frac{100+Ws*C}{100} \qquad (2)$$

where $P_{cor}$ (mm) is the corrected precipitation, $P_{adj}$ (mm) is the measured precipitation after filtering, CE is the catch efficiency of the Geonor, Ws (m s$^{-1}$) is the hourly mean wind speed at the gauge height, C is a constant and represents the gauge configuration parameter and is equal to –7.1 for the single Alter-shield Geonor (Thériault et al., 2012). Over-

correction is possible for snowfall events and occurs under the impact of blowing snow at high wind speeds (Pan et al. 2016). To limit over-correction, we used a threshold for maximum wind speed. Thus, the CE was set to 0.44 (CE for 8 m s$^{-1}$). We used this threshold since the median collection efficiency of the Geonor in the single Alter-shield seemed to saturate at wind speed greater than 8 m s$^{-1}$ for a test site in Colorado (Thériault et al., 2012). We adjusted all rainfall measurements using the constant average catch efficiency for the single Alter-shield Geonor estimated at 95% (CE = 0.953) (Devine and

Mekis, 2008).

Wind speed at gauge height is required for the estimation of CE. We used the logarithmic wind profile to estimate the wind speed at the height of the Geonor gauge (Yang et al. 1998):

$$W_s(h) = W_s(H) \left[ \frac{\ln(h/z_0)}{\ln(H/z_0)} \right] \qquad (3)$$

where Ws(h) is the estimated hourly wind speed (m s$^{-1}$) at the gauge height, Ws(H) is the measured mean hourly wind speed at the anemometer height, h and H are the heights (m) of the Geonor gauge and the anemometer respectively, $z_0$ is the



roughness parameter (m) set to 0.01 m for the winter snow surface and 0.03 m for short grass in the warm period (Yang et al. 1998). For this study, we set z0 = 0.01 for the time period between December and March and $z_0$ = 0.03 for the rest.

Different approaches for precipitation phase determination has been summarized in Harpold et al., (2017). Most common methods rely on the use of mean air temperature (Ta) thresholds (e.g., Marks et al., 2013). In this study, we use a static

threshold of 0°C (Marks et al., 2013) to distinguish between snowfall and rainfall. All precipitation below or above the threshold are partitioned as snow or rain, respectively. This method was found to yield reliable snow precipitation in Idaho when cloud levels are at or close to the surface during storms and the RH is at or close to saturation (Marks et al., 2013). The Mount-Lebanon meteorological conditions during storm events, are similar, and are usually characterized by RH saturation and cloud levels are near the surface.

**3.3 Snow course data**

We identified 30 different snow courses with lengths varying between 75 and 400 m, within the upper area of the three basins (Fig. 1). The locations of the snow courses were selected based on accessibility, representativeness of the snow cover within the region (suggesting, whenever possible, 1 snow course for relatively flat and low slope regions and 2 snow courses representing the maximum and minimum snow depth transects in rough topographic regions). Snow courses were spaced at

~100 m vertical elevation. All snow courses had a slope of less than 30%. Field measurements of snow depth, snow density, and were carried over two snow seasons (2014–2016) with an average revisit time of 11.4 days for each snow course. A total of 649 snow course measurements are reported and can be found at (doi.org/10.5281/zenodo.321405). Snow depth was measured manually, to the nearest 1 cm using a 3 meter snow probe, at 5 meter interval along each snow course. Snow density was measured using federal snow sampler (snow cutter inner diameter at 3.772 cm) along each snow course at 25

meter intervals for snow courses shorter than 100 m and at 50 m intervals for longer snow courses so at least 3–5 snow density measurements were recorded at each snow course site. Snow density and SWE protocol consisted of weighting the empty tube. The snow tube was then plunged into the snowpack and the snow depth marker on the tube was recorder. Once the core was removed the snow depth of the snowpack was checked to make sure that at least 80% of the snowpack has been cored (Dixon and Boon, 2012). We also measured the snowpack HS using a marked snow probe (1 cm) to make sure no

snow was left unsampled. Any amount of soil entering the snow cutter, especially during melt season, was removed and reported alongside the HS measurements. The combined weight of the tube and core was recorded. Each snow sample was weighted 3 times (5 times under windy conditions) and the average snow weight was registered. Under windy conditions maximum and minimum weights were removed and the weigh was averaged for the three measurements. The mass of the snow core sample was calculated by subtracting the empty tube mass from the combined tube and snow core mass. Snow

courses with an HS of less than 30 cm and where the snow cover was less than 50 % (based on visual interpretation) were sampled by taking bulk density measurements. Bulk density measurement consisted on taking 4 snow core samples at a single location and weighting the total mass for the combined 4 samples, the average density is then reported for this point-location. Weighting scales were validated under normal weather conditions by taking 50 measures of the empty snow tube



and thus suggesting an accuracy of 98.75% when using a 2 m snow tube. Snow density and snow water equivalent were calculated using (Eq. 4 and Eq. 5):

$$\rho_S = \frac{m_{sample}}{\pi r^2 \times SD} \tag{4}$$

$$SWE = SD\frac{\rho_s}{\rho_w} \tag{5}$$

where $\rho_s$ is the density of the snow core sample (g cm$^{-3}$), $\rho_w$ is the density of water (1 g cm$^{-3}$), $m_{sample}$ is the mass of the snow core sample, r is the inside radius of the snow tube cutter (3.772 cm), and HS is snow height (cm).

## 3.4 Snow cover extent and snow cover data

Daily maps of the snow cover extent at 500 m spatial resolution were generated for the three watersheds from the MODIS snow products. We used the "binary" snow cover area sub–dataset from MOD10A1 (Terra) and MYD10A1 (Aqua) collection 5 products from the National Snow and Ice Data Center (Hall et al., 2006). Mount Lebanon falls between MODIS grid tiles h20v05 and h21v05. All available tiles from 01 September 2011 to 31 August 2016 were assembled and resampled to 500 m with a nearest neighbor method in UTM 36N using the MODIS reprojection tool over a rectangular spatial subset (upper left x=730 km; y=3830 km, lower right 850 km 3680 km). Then we ran a gap–filling code which is fully described in (Gascoin et al., 2015) to interpolate the missing information mostly caused by cloud cover. The algorithm utilizes the topographic information (elevation and aspect) from the ASTER GDEM, which was resampled to the same resolution. The output is a series of daily, gap–free, raster maps of the snow presence or absence for every pixel. These data were then used to compute the daily snow cover area in each watershed and the mean annual snow cover duration per pixel (SCD), i.e. the mean number of snow days per year (Fig. 9). The dataset is available at (doi.org/10.5281/zenodo.321405).

## 4 Results and discussion

In this section, we limited the data analysis to two snow seasons (2014–2016) because (1) the AWS network became fully operational with the installation of the third station in 2014 and (2) snow field observations were collected starting the snow season 2014–2015.

## 4.1 Meteorology

The observed average seasonal (average 30–min from 01 November to 30 June) surface air temperature for the two snow seasons (2014–2016) was 6.93, 4.26, and -1.36 °C for LAQ (monthly range: -0.3–17.1 °C), MZA (-2.9–14.3 °C), and CED ( -7.1–8.4 °C) respectively. There is a strong positive correlation in the 30 min surface air temperature records from the three stations (r = 92.7–97.9). Total annual precipitation ranged between 732 and 1125 mm during 2014–2015 and 1592 to 1880



mm for 2015–2016 representing precipitation data recorder at the LAQ and MZA stations respectively (1840–2294 m a.s.l.). Average wind speed for the same time period (2014–2016) were 2.44 m s$^{-1}$ (monthly range: 1.5–3.3 m s$^{-1}$), 5.35 (4.1–8.0 m s$^{-1}$), and 4.73 (3.6–5.5 m s$^{-1}$) m s$^{-1}$ for LAQ, MZA, and CED respectively. Strong winds seldom exceed the 10 m s$^{-1}$ (LAQ), 20 m s$^{-1}$ at MZA, and 25 m s$^{-1}$ (CED) except during storm events were maximum wind gusts recorded reached up to 40.1 m s$^{-1}$ at CED. Seasonal incoming solar shortwave radiation averages (30–min averages) ranged between 156–219 W m$^{-2}$ for the three stations (2014–2016).

## 4.2 Snow depth, snow density and SWE

SWE peaks in mid–February at low and mid–altitude regions and in mid–March high mountainous regions. Snowmelt varies depending on elevation and begins late February at lower altitudes and by mid–March at higher altitudes and extends into late April. Rain on snow event is common during winter season and usually occur at elevations below 1800 m amsl. Snow patches can persist until June in areas above 2700 m a.s.l.. The median HS, SWE, and density obtained from snow courses (1300–2900 m a.s.l.) between 2014 and 2016 are shown in Fig. 6. The median HS, SWE, and density across different region and mountain elevation are illustrated in (Fig. 7). High regions like CED (1650–2900) and MZA (1300–2300) have higher mean HS and SWE when compared to mid and low mountainous regions like LAQ (1300–1850). Median seasonal HS values at high elevations (i.e. mean for all CED and MZA snow courses; Fig. 7a,d) were very close which reflects similar snowfall patterns. CED seasonal HS medians were 77 and 79 cm representing snow years 2014–2015 and 2015–2016 respectively and similar values were found in MZA (71 and 76 cm). Meanwhile, HS medians at LAQ were 53 and 54 cm over the same time period. The high mountainous regions above 2200 (e.g., CED) have a higher 75th quantile range and this is attributed to the extend snow persistence. The higher maximum HS observed in MZA can be attributed to the rough topography of the region whereas the high region in CED above 2700 m a.s.l. is presented as a plateau with less variance in HS and SWE when compared to high regions in MZA (2100–2500).

Peak SWE values for the two winter seasons (2014–2015; 2015–2016) were 103 and 83 cm w.e. for CED; 127 and 158 cm w.e. for MZA, and 59 and 36 cm w.e. for LAQ. The peaks for SWE, and similarly for HS, observed at MZA (Fig. 7) are attributed to the topography of the region were wind–blown snow was more noticeable. Despite MZA is characterized with higher variability in the SWE and HS, observations at CED shows higher seasonal medians for both variables and this expected since CED is higher and snow season extends longer than the one observed at MZA. SWE remains relatively constant during winter season and starts to melt starting Mid–March or early–April at regions above 2100 m a.s.l. Snowmelt at lower elevation 1600–2100 m a.s.l. started earlier by mid–February and early March. Regions between 1300 and 2000 m a.s.l. are subject to rain on snow which influence snow melt processes during the entire snow season especially in warm years. The higher variance and inter–seasonal variability in the observed SWE across the different regions (Fig. 5) illustrate the importance for monitoring snowpack dynamics in Mount–Lebanon as it is the major contributor to the water resources system.



The median seasonal snow density for all snow courses over the two years' period (2014–2016) (Fig. 6) was 476 km m$^{-3}$ and ranged between 431 and 522 kg m$^{-3}$ representing the first 25% and last 75%. Mean snow density for the three regions (Fig. 7c,f) over the two snow seasons (2014–2016) ranged between 440 and 489 kg m$^{-3}$. These high seasonal density values are common in Mediterranean regions (e.g., Rice and Bales, 2010; López–Moreno et al., 2013). Although close median

values can be observed for snow density across the different regions it is good to note that different snow metamorphism, compaction, or melting differences exist between regions. During the months of January and February the Mid–altitude regions (e.g., LAQ) snow is usually wetter and less supportable when compared to high elevation regions (e.g., CED) where snow is usually wind compacted, highly supportable and dry.

Snow year 2015–2016 was characterized with rain on snow events when most of the precipitation occurring few days

after the first major snowfall event (mid–January) felt as rain at mid–altitude regions (1300–2300 m a.s.l.). Rain on snow resulted in the disappearance of the snowpack below 1800 m a.s.l and accelerated the snow densification at elevations between 1800 and 2300 m a.s.l. Observed snow densities for the same date were 1.4–1.6 times higher than those observed during the same time–period in the previous year (2014–2015).

### 4.3 Modelling snow bulk density

Establishing the relationship between distributed snow depth and SWE is among the used approaches to quantify SWE from snow depth measurements (e.g., Jonas et al., 2009). Fig. 8. shows the general relationship between density vs HS (Fig. 8a) and SWE vs HS (Fig. 8b). The higher densities (Fig. 8b) are typical with Mediterranean, warm maritime, and alpine regions (e.g., Sturm et al. 2010). In contrast, the observed scatter between snow density and HS (Fig. 8b) cannot be explained using linear estimators. While SWE can be estimated linearly from HS, it is recommended to model the bulk density from HS and

then derive SWE (Sturm et al. 2010). Such approach is justified by the facts that (1) depth varies over a range that is many times greater than that of bulk density and (2) because estimates derived from measured depths and modelled densities are usually very close to measured values of SWE. The snow density can be estimated from HS using a linear function. However, a better representation of snow density from snow depth measurement can be achieved using nonlinear function that include HS and account for the effect of snow aging (represented in term of day of the year (DOY)) (Sturm et al. 2010).

Distinct class for different climate regions are used to account, indirectly, for the effects of meteorological condition (i.e. temperature and wind) (Sturm et al., 2010). The general equation is a nonlinear function asymptotic to the maximum seasonal density (Sturm et al. 2010) (Eq. 6):

$$\rho_{h_i,DOY_i} = (\rho_{max} - \rho_0)\left[1 - e^{(-k_1 \times h_i - k_2 \times DOY_i)}\right] + \rho_0 \qquad (6)$$

where $\rho_{max}$ and $\rho_0$ are maximum and minimum bulk density, k1 and k2 are densification parameters and hi is snow depth at the ith observation. $\rho_{max}$, $\rho_0$, $k_1$, and $k_2$ vary with climate region and the model parameters for the major snow class are found in Sturm et al. (2010) Table 4. The equation was applied to the ensemble points, presented in Fig. 8b, using snow depth and

DOY as predictor variables and snow depth as predictand. The model parameters were $\rho_{max} = 0.551$, $\rho_0 = 0.0369$, $k_1 = -0.0019$, $k_2 = 0.0179$ for the entire dataset with the model explaining 34% (coefficient of determination $r^2 = 0.344$) of the snow density variability (Table 4). We believe that some of the observed differences between the current Mediterranean and the maritime and alpine regions in general can be attributed to the shorter snow season, warmer temperature, and higher densification rates especially in the mid–elevation zone (1300–1900 m a.s.l.). During February field visits, most of the observed snow at this mid–elevation zone was wetter and characterized with higher densities when compared to high elevation regions (e.g., above 2200 m a.s.l. where the snow was usually dry). Better snow density fit was achieved using elevation bands with a better fit for elevation above 2200 m (Table 4). The model was not able to explain the variability of snow densities to snow elevation in low mountain regions (1300–1800 m a.s.l.) namely because this region is subject to rain on snow events and multiple snow accumulation and melt during a single snow season.

## 4.4 Remote sensing snow cover data

MODIS data indicate that snow falls occurred between November and March. The snow cover area peaked between January and February. Maximum snow cover duration (SCD) was 160 days at higher altitudes (above 2700 m a.s.l.). SCD in medium elevation mountain regions (2200–2600 m a.s.l.) ranged between 100 and 140 days per average year. The percent of snow covered area (SCA) of the basins during winter months (December – March) ranged between 28–46% (Abou Ali), 36–66% (Ibrahim) and 27–50% (El Kelb) (Fig. 9).

## 5 Data availability

All data described in this paper are made publicly available at Zenodo (doi.org/10.5281/zenodo.321405). Included are comma–separated files (.csv) for AWS listing the three stations, snow course observation, and a compressed file for processed daily MODIS SCA and SCD.

## 6 Conclusions

This paper presents the first dataset of snow and meteorological conditions in Mount–Lebanon. The observations focused on three major basins of the coastal region of Lebanon and cover the snow seasons between 2011 and 2016 (for MZA AWS located at 2300 m a.s.l.). The observation network became fully operational in 2014. The network includes three automatic stations covering the range of snow dominated areas between 1840 and 2834 m a.s.l. Distributed in-situ HS and SWE measurements were also collected during two snow seasons (2014–2016) at 30 different snow courses located between 1300 and 2900 m a.s.l.. MODIS snow products were processed to compute SCA and SCD at the basin scale. These observations are the result of an ongoing joint–collaboration between IRD (France), CESBIO (France), CNRS/NCRS (Lebanon), and USJ (Lebanon). The observatory is currently being funded for two years (2016–2018) via grants from CNRS/NCRS, IRD and USJ.



Additional unpublished data observations may also be available to complete this dataset, including meteorological data collected by the meteorological services at the department of civil aviation and the Lebanese agricultural research institute (LARI), and river and spring discharges monitored by the Litani river authority (LRA) as well as meteorological and hydrological data combined by the early warning system at the national center for remote sensing (NCRS) and the

observational datasets at the Centre d'Information et de Formation aux Métiers de l'Eau (CIFME) at the Ministry of Energy and Water (MOEW).

We provided mean and seasonal snow properties over two snow seasons (2014–2016) and provided an example on how SWE and snow density can be obtained using only HS measurements. This time span is insufficient to characterize the temporal variability of the snow cover. However, it already provides consistent information on the snow spatial variability.

The combination of the meteorological station, snow courses, and remote sensing data through the application of a snowpack model will enable a multi–year evaluation of the snow resources at the basin scales.

The accurate representation of the spatial distribution of HS, SWE, and snow density is crucial for hydrological applications. In particular, we are using the AWS data are for running a distributed energy balance model. The snow observations also hold potentials for the characterization of the spatial distribution of snow across different gradients. SCA

and SCD data can be used for model validation or as an operational tool for water resources management (e.g., Sproles et al. 2016). The AWS data may also support the validation and downscaling of regional climate models for various applications beyond the study of snow hydrology and the use of water resources.

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

Faraya and Mzar between 1350 and 2350 m; and 6 snow courses between Ehmej and Laqlouq (elevation range between
1350 and 1850 m).



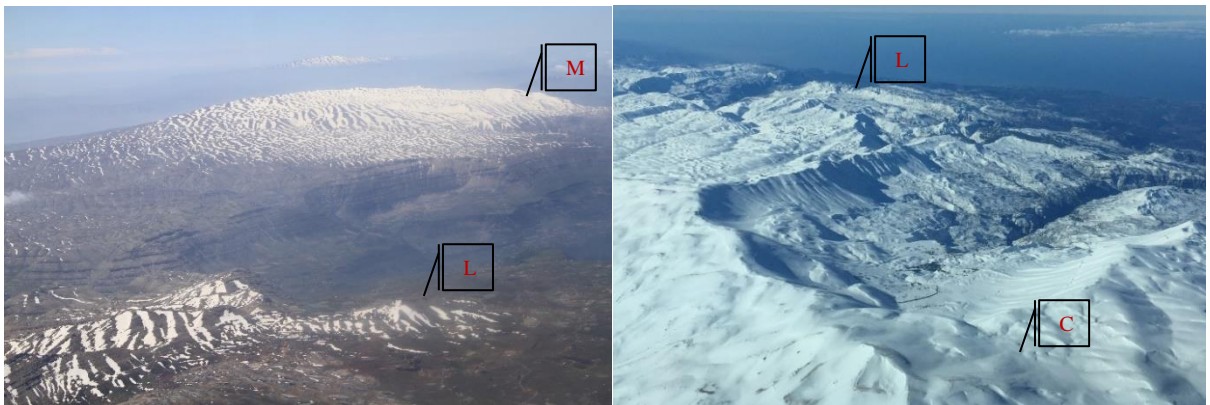

**Figure 2:** Overview of the mountain topography at (a) Laqlouq and Mzar, and (b) Cedars, Mount–Lebanon. The images were captured on 6 May 2011 (a) and 21 February 2015 (b) (courtesy of the author). The locations of the AWS are show approximately on the images where the letters M, L, C represents the stations at MZA, LAQ and CED respectively (see
20  Table 1). The topography near Laqlouq (LAQ) is relatively low plain (elevation between 1600 and 1800 m a.s.l.) and low elevation mountain (1900 – 2100 m a.s.l.). The region near Mzar (MZA) is characterized with rugged terrain (1600 – 2200 m a.s.l.) and mid-elevation plateau (elevation range 2300 – 2500 m). The high elevation plateau (shown partially in (b) near Cedars (CED) have an elevation range between 2700 and 3000 m a.s.l. Snow persistence lasts till the end of May in the mid-elevation mountain regions (plateau and rugged terrain region above 2300 m a.s.l.). The low elevation and mid-elevation
25  regions (1300 – 2000 m a.s.l.) are usually snow free by Mid-March to Mid-April.



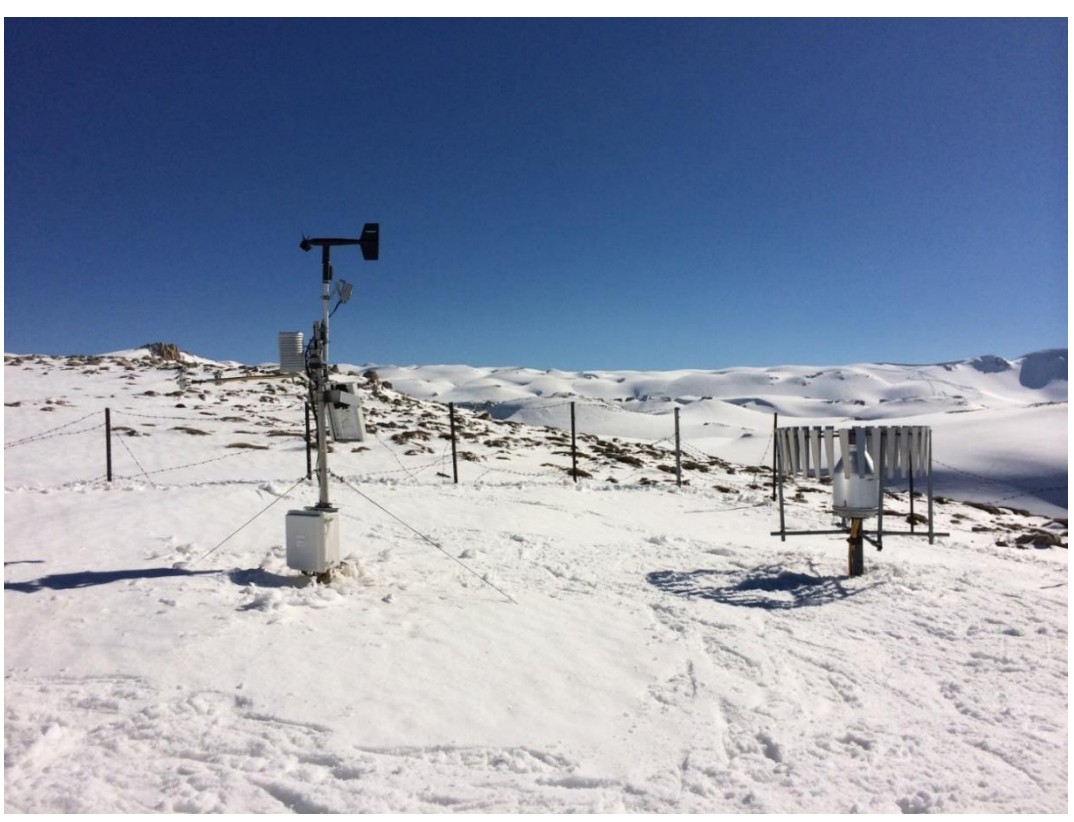

**Figure 3:** Automatic weather station at Mzar (MZA) (2296 m a.s.l.) where all sensors are located on the tower and the
10   precipitation gauge is located to the right of the station. Image captured on March 5th, 2015 (courtesy of the author).





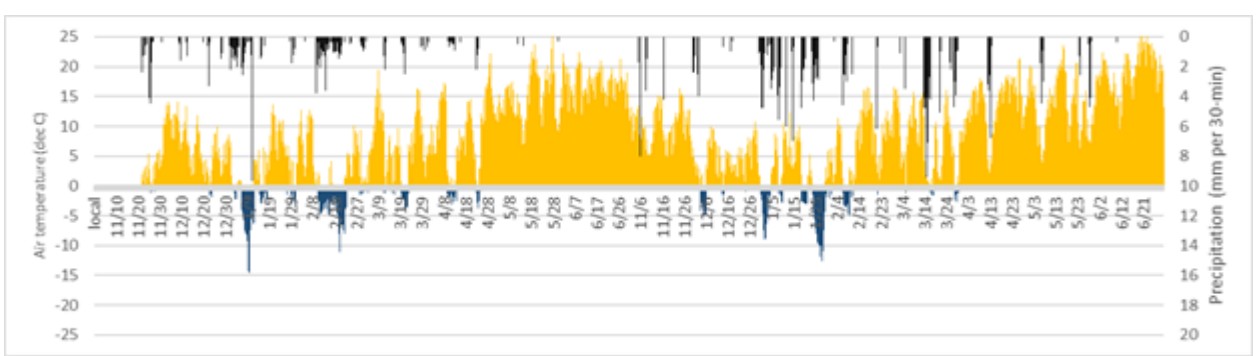

**Figure 4:** Example of hourly precipitation and temperature observations at Laqlouq (December–June, 2014–2016) (1840 m a.s.l.).




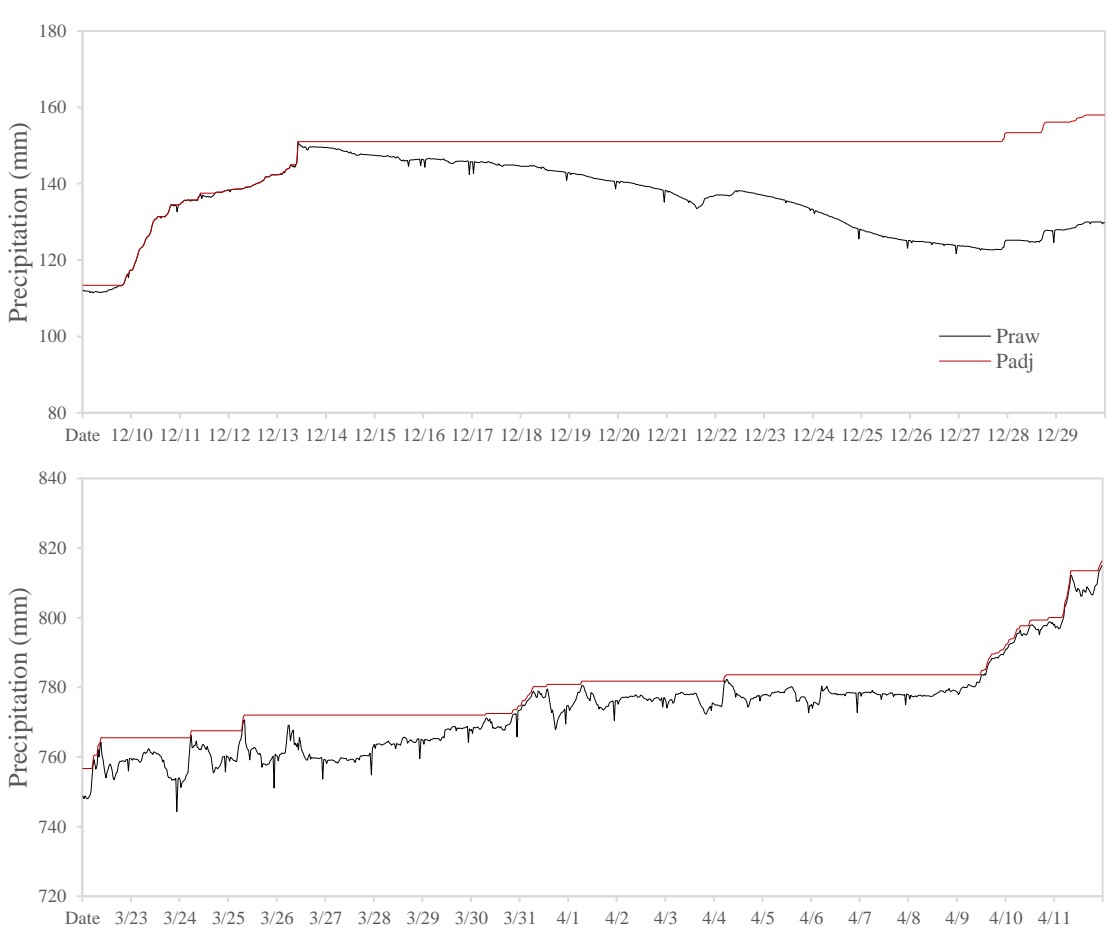

**Figure 5:** Examples of the jitters and diurnal noise filtering for Geonor T-200B weighing gauge (Praw: raw data; Padj: filtered data). (a) Significant evaporation occurred during winter season 2014 (e.g., MZA: 12 – 29 December, 2013) and required manual correction (28 – 29 December) no correction for the accumulation of raw precipitation between 22 – and 23 December was made because the observed average humidity was below 15%. (b) Filtering of jitters and diurnal noise (no manual correction) (e.g., MZA: 23 March – 11 April, 2015).





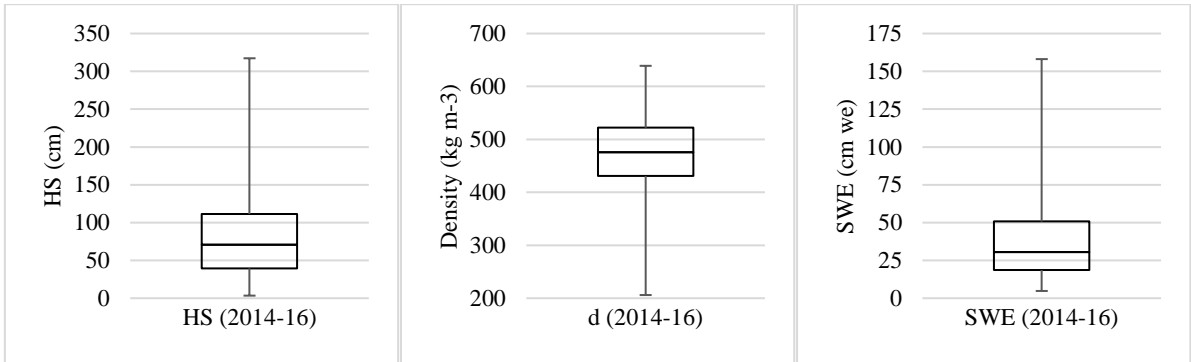

**Figure 6:** Box and whisker plots for non–zero data where (a) snow depth, (b) snow density, and (c) SWE over the two snow seasons 2014–2016 using data from 30 snow courses located at elevations between 1300 and 2900 m a.s.l (n = 649). The box
brackets are 25% and 75% of the data (lower and upper boxes respectively). The whiskers are at minimum and maximum values.





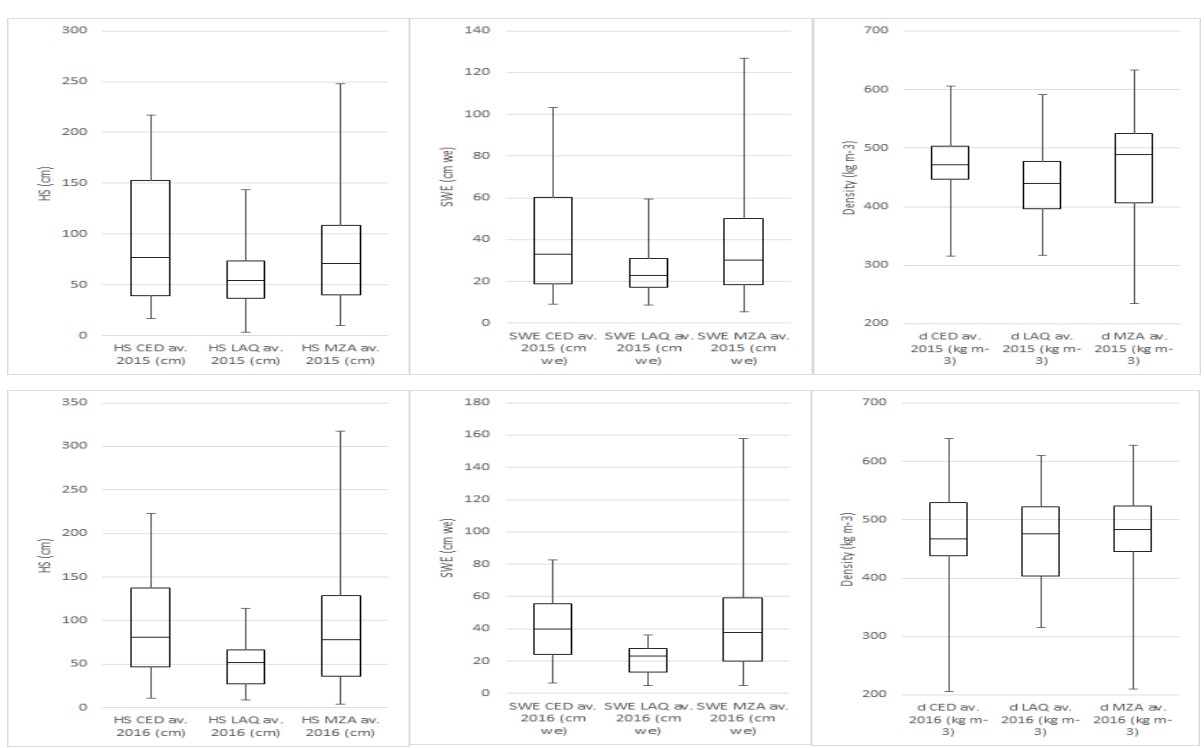

**Figure 7:** Box and whisker plots for non–zero data (a) snow depth, (b) snow density, and (c) SWE over two snow season 2014–2015 (top; n = 311) and 2015–2016(bottom; n = 371) distributed by region where CED (elevation range, 1650–2900 m), LAQ (1300 – 1850 m) and MZA (1350 – 2350 m). The box brackets are 25% and 75% of the data (lower and upper boxes respectively). The whiskers are at minimum and maximum values.



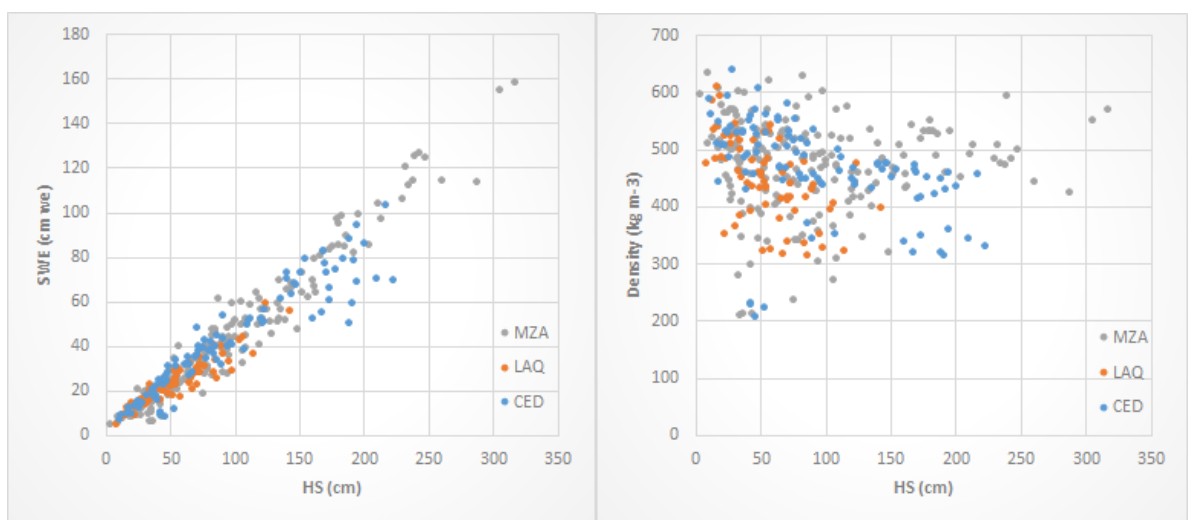

**Figure 8:** (a) SWE vs depth and (b) density vs depth for all snow courses data (observed during two snow seasons 2014–2016 at elevations between 1300 and 2900 m a.s.l.) (n = 649).



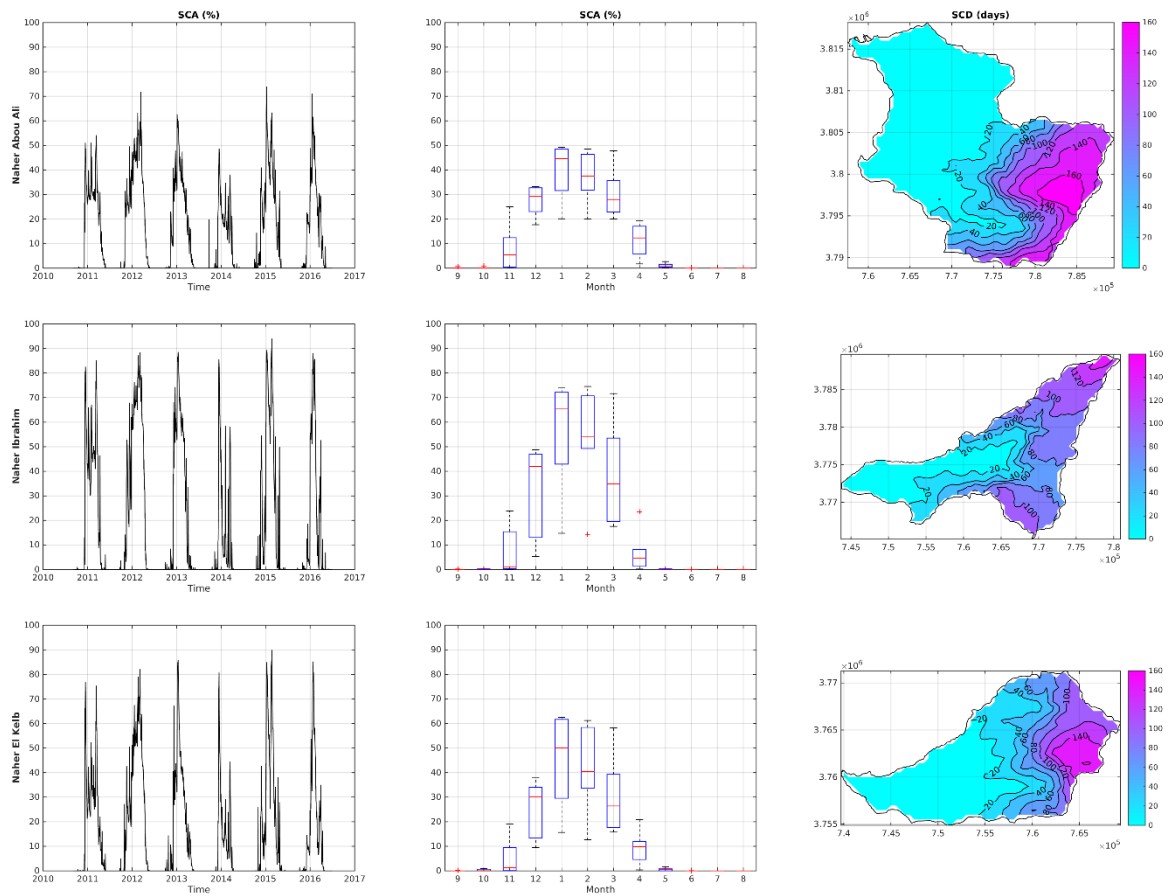

**Figure 9:** From left to right: time series of the snow cover area (SCA, percentage of the basin area), boxplot of mean monthly SCA, and SCD for (from top to bottom) Abou Ali, Ibrahim, and El Kelb River Basins.





**Table 1:** Attributes of the three snow-dominated basins in Mount-Lebanon described in this study.

| Basin[a] | Area (km$^2$) | Elevation range (average)[b], m a.s.l. | Dominant land cover[c] (%) | AWS Elevation, m a.s.l. (year installed)[d] | Snow courses count (elevation range[e] |
|---|---|---|---|---|---|
| 1 | 513 | 0–3088 (1202) | Clear grassland (20%) | 2834 (2013) | 9 (1650–2900) |
| 2 | 323 | 0–2681 (1547) | Clear grassland (30%) | 1840 (2014) | 6 (1300–1850) |
| 3 | 256 | 0–2619 (1381) | Clear grassland (16%) | 2296 (2011) | 15 (1300–2300) |

[a]Basins are Abou Ali (1), Ibrahim (2), and El Kelb (3) (Fig. 1)
[b]Values are derived for the national 10 meter DEM (NCRS)
[c]Source: Landuse land cover map of Lebanon (NCRS, 2015)
[d]Source: Institut de recherche pour le développement (IRD) (Tabel 2)
[e]Snow courses observations were conducted between December and May over two snow years (2014–2016) (Fig. 1).



**Table 2:** Meteorological stations.

| Station[*] | Location | Elevation, m a.s.l. | Record period (30 min–averages)[†] | Coordinates (WGS84) |
|---|---|---|---|---|
| CED | Cedars | 2834 | 2013–2016 | 34.27N; 36.09E |
| LAQ | Laqlouq | 1840 | 2014–2016 | 34.14N; 35.88E |
| MZA | Mzar | 2296 | 2011–2016 | 33.98N; 35.86E |

[*]See Table 3 for sensors description. [†]The time period cover the snow season between 1 November and 30 June.



**Table 3:** Sensors specifications and quality control checks for semi-hourly and daily data – adopted after Estévez et al. (2011) and WMO (2008).

| Sensor | Variable | Accuracy (Sensibility) | Range test | Step test | Cross–validation test |
|---|---|---|---|---|---|
| T–200B (1000mm at MZA and LAQ) and 1500mm at CED†) | Precipitation (mm) | 0.1% Full scale (0.075 – 0.1) | $0 \leq Psh \leq 120$ $0 \leq P \leq 508$ | $0 \leq Psh$ ; $P(0 - 6h) \leq P(0 - 24h)$; | RHsh > 80% |
| SR50 | Snow depth (cm) | ±1 (0.25) | $0 \leq SDsh \leq 450$ | | Maximum SD(0–24h) < 0.15*P(0–24h) |
| CS215 | Temperature (°C) | ±0.4 | $-30 < T < 50$ | $|Tsh - Tsh_{-1}| < 3$ | $Tsh \neq Tsh_{-1} \neq Tsh_{-2} \neq Tsh_{-4}$ |
| SP LITE 2 | Incoming Radiation | | $-1 < SwIsh < 1500$ | $0 \leq |SwIsh - SwRsh_{-1}| \leq 555$ | For SwIsh > 0 & SwRsh > 0 [0 < Albedo (SwR/SwI) < 0.95] |
| | Reflected Radiation | | $-1 < SwRsh < 1500$ | $0 \leq SwRsh - SwRsh_{-1} \leq 550$ | |
| CS215 | Relative Humidity (%) | ±0.2 | $0.8 < RH < 103$ | $|RHsh - RHsh_{-1}| \leq 45$ | |
| Young 05103 – Alpine | Wind Speed (m s$^{-1}$) | ±0.3 | $0 < Ws < 60$ | | WSsh = 0 & WDsh = 0; WSsh ≠ WSsh$_{-1}$ WSsh ≠ WSsh$_{-2}$ ≠ WSsh$_{-4}$; WDsh ≠ WDsh$_{-1}$ WDsh ≠ WDsh$_{-2}$ ≠ WDsh$_{-4}$ |
| | Wind direction (deg) | ±3 | $0 \leq Wd \leq 360$ | | |

Where: P and Psh = daily and semi-hourly precipitation; SD and SDsh = daily and semi-hourly snow depth; T = mean air temperature respectively; Tsh = semi-hourly air temperature; SwI and SwR = incoming and reflected solar radiation respectively (sh denotes semi-hourly); RH and RHsh = daily mean and semi-hourly relative humidity; $W_S$ and Wd = mean wind speed and mean wind direction respectively (sh denotes semi-hourly); For SD we used visual interpretation to account for snow depth following snowfall, or when SR50 measurement are lost, assuming the difference in SD over a single day is less than total daily precipitation multiplied by an average fresh snow density of 0.15 g cm$^{-1}$. The sensor's field life cycle is ~ 3 years. †The snow gauge (model T–200B, 1500mm) was installed in October 2016 and measurements will be available staring snow year 2016–2017).





15 **Table 4:** Model parameters by elevation bands for non–zero data.

| Elevation range (m a.s.l.) | $\rho_{max}$ | $\rho_0$ | $k_1$ | $k_2$ | $r^2$ |
|---|---|---|---|---|---|
| 2200–2900 (n = 136) | 0.582 | 0.229 | 0.0004 | 0.0139 | 0.616 |
| 1300–2900 (n = 353) | 0.553 | 0.345 | 0.0000 | 0.0167 | 0.344 |