# Peer review of "Snow observations in Mount-Lebanon (2011-2016)"

_Earth System Science Data, 2017_

## Referee Comment (RC1) · Anonymous Referee #1 · 5 Apr 2017

The authors present a meteorological and snow dataset collected over 2011-2016 in Mount-Lebanon. This includes 30-min meteorological observations at three sites, snow course measurements of snow depth, SWE, and density at 30 locations, and 500-m snow covered area and snow covered duration maps from MODIS. Given the lack of snow data in this region of the world, the paper has the potential to make a novel contribution. I could envision these sites being used in a snow model intercomparison study, which are often conducted across diverse sites. That being said, there are many major aspects that require the authors' attention before this should be considered for publication in ESSD.

MAJOR COMMENTS

- In the provided dataset, there are periods when the outgoing shortwave data exceed

the incoming shortwave data. This may occur during periods when snow is covering the up-pointing pyranometer but the down-pointing pyranometer is snow-free (e.g., see Lapo et al. 2015). While the text suggests that the incoming and reflected shortwave radiation measurements were screened by constraining albedo between 0 and 1, the provided data do not support this claim. The attached figure shows the problem. This needs further attention.

- The provided snow course data exhibit a fundamental inconsistency. The bulk snow density (with respect to water density) is the ratio of SWE by snow depth (as in Equation 5). However, when I compute this value from SWE and snow depth and compare to the density values in the dataset, they do not match (see attached figure). Please revisit the data and correct this issue. Because figures and analysis revolves around these data, it is essential to rectify these problems.

- The sites do not have meteorological data that are gap-free and consequently are not immediately useful to modelers and others looking for sites to run/test models. The convention of many other snow data papers has been to provide data that are complete in time, using various in-filling techniques. The data presented here would be of greater value if the gaps were filled.

GENERAL COMMENTS

- Are the temperature sensors naturally or mechanically ventilated? Please note this in section 3, perhaps in the paragraph on page 4, starting at line 25. To enhance the value of your dataset, you could consider preparing a corrected temperature dataset based on your reflected shortwave dataset based on Huwald et al. (2009).

- While the sites do provide valuable meteorological and snow data in a unique environment, I would advise the authors to note that the sites do not measure incoming longwave radiation, a key variable for the energy balance of warm, Mediterranean climates.

[Figure]

- Please include the elevation and coordinate (e.g. latitude/longitude) data for each of the 30 snow course sites.

- Some of the figures are difficult to read because of low resolution or text fonts that are too small. These include Figures 4, 7, and 9.

- In numerous places (e.g., P.7, L.21), the authors use the word "weighting" when they should actually be using the word "weighing" (no "t"). These words have different meanings.

- The procedures for measuring snow depth, SWE, and bulk density with the federal sampler (P. 7, L.19-26) are fairly standard, so this section may be providing too much detail and can be greatly reduced.

TECHNICAL CORRECTIONS

- P.1, L.15: It should read "Precipitation data were" (data are plural).

- P.2, L.20: Add "an" before "operational snow observation network".

- P.2, L.24: Replace "were made" with "are".

- P.2, L.32: This is the first usage of the acronym "AWS" in the paper, but it has not been defined yet. Please define.

- P.3, L.12: Replace "covers" with "and covering".

- P.4, L.15-16: The sentence is confusing. It can mean that each of the sites have temperature/humidity sensors at three heights above ground, or that these are the heights at the three sites, respectively. I think it is the second meaning. Please rephrase for clarity.

- P.4, L.19: Replace "is being observed" with "has been observed".

- P.5, L.14: Replace "was adjusted" with "were adjusted" (data are plural).

- P.5, L.17: Replace "was removed" with "were "removed" (changes=plural).
- P.5, L.22: Replace "is greater" with "was greater" (past tense).

- P.5, L.27: Add "event" after "precipitation".

- P.5, L.31-32: Replace "is" with "was" before "equal", before "assumed", and before "preserved" (past tense).

- P.7, L.2: Please use a subscript for the zero on "z0" (first case on this line).

- P.7, L.16: I think you might need to add "SWE" before "were carried"?

- P.7, L.19: Add "a" before "federal snow sampler".

- P.7, L.22: Replace "recorder" with "recorded".

- P.8, L.21: Please remove the reference to Fig. 9 here. At this point, Figures 6-8 have not been introduced, so it is confusing to reference Figure 9 before 6-8.

- P.8, L.24-25: Please replace "starting the snow season 2014-2015" with "starting in snow season 2014-2015".

- P.8, L.28: Considering the measurement precision of the temperature sensors, I think it is appropriate to only reference to the nearest tenth of a degree here (and elsewhere). Please replace "6.93, 4.26, and -1.36" with "6.9, 4.3, and -1.4". The same comment holds for the wind speed averages (P.9, L.2-3).

- P.9, L.1: Replace "recorder" with "recorded".

- P.9, L.10: Replace "Rain on snow event is" with "Rain on snow events are".

- P.10, L.7: The word "supportable" is vague in this sentence. Please select a different word or phrasing.

- P.10, L.9: Add "a" before "few days".

- P.10, L.10: Change "felt" to "fell".

- P.10, L.16-17: These are backwards. Fig. 8a shows HS vs. SWE while Fig. 8b shows

HS vs. density. Please correct.

- P.10, L.20: Add "an" before "approach".

- P.10, L.23: Add "a" before "nonlinear".

- P.10, L.24: Change from "account" to "accounts".

- P.10, L.30: Please use a subscript for "i" in "hi".

- P.11, L.12: Replace "snow falls " with "snow storm events".

- P.12, L.13: Delete "are" after "AWS data".

- P.12, L.14: Change "potentials" to "potential".

- P.16, L.18: Replace "show" with "shown".

- P.16, L.23: Replace "till" with "until" to avoid slang.

- P.19, L.12: Start a new sentence after the parentheses, i.e.: "No correction for the accumulation...".

- P.24, L.21: Correct the spelling error. Replace "Tabel" with "Table".

TABLE AND FIGURE COMMENTS - Figure 2: Please add "(a)" and "(b)" to the panels of this figure.

- Figure 4: For clarity, please add either a legend to differentiate the different datasets or a note in the caption to identify which data pair with each vertical axis.

- Figure 5: Please add "(a)" and "(b)" to the panels of this figure.

- Figure 6: Please add "(a)","(b)", and "(c)" to the panels of this figure.

- Figure 7: Please add "(a)","(b)", and "(c)" to the panels of this figure. You may need to include "(d)", "(e)", and "(f)" as these are referenced in the text.

- Figure 7: The caption says that (b) is the snow density, but the middle plots show

[Figure]

SWE. Please correct either the caption or the ordering of the panels.

REFERENCES

Huwald, H., C. W. Higgins, M.-O. Boldi, E. Bou-Zeid, M. Lehning, and M. B. Parlange (2009), Albedo effect on radiative errors in air temperature measurements, Water Resour. Res., 45(8), W08431, doi:10.1029/2008WR007600.

Lapo, K. E., L. M. Hinkelman, C. C. Landry, A. K. Massmann, and J. D. Lundquist (2015), A simple algorithm for identifying periods of snow accumulation on a radiometer, Water Resour. Res., 51(9), 7820–7828, doi:10.1002/2015WR017590.

**Fig. 1.** Comparison of incoming and reflected shortwave data at the sites

[Figure]

**Fig. 2.** Comparison of provided and computed snow density

[Figure]

---

## Short Comment (SC1) · 11 Apr 2017

We greatly appreciate the referee's constructive comments, which will help us improve our manuscript. We will provide a detailed response to every comment once we have received the next reviews, but here we would like to respond briefly regarding the issues in the AWS radiation data and snow course data that were raised by the referee.

1) Thank you very much for pointing us that we provided the raw incoming and outgoing solar radiation data by mistake, instead of the processed data as indicated in the manuscript. The radiation data were flagged using the following classes (0: zero incoming and reflected shortwave radiation measurements; 1: data are correct, albedo test is OK (is between 0 and 1); 2: data do not pass the albedo test; 3: observation bias likely due to frost on the sensor; 4: field observations reported that the sensor was not well levelled; and 5: no data). Extracting shortwave data with flag 1 only results in albedo values that are within 0 and 1. We also used visual inspection to check for potential biases arising from snow frost especially when the outgoing shortwave exceeded the incoming SW during winter season. The corrected data is shown in Fig. 1. and can be accessed following this temporal link (https://www.dropbox.com/sh/zs9klzhl55ezdss/AAB-0wgKfHGheMmE05ovmFfpa?dl=0). The data in the public repository will be upgraded with the revised version.

2) The apparent inconsistencies in the snow height (HS), snow water equivalent and density are due to different sampling strategies during a snow course. We provided HS that is the average of the HS measurements collected at 5 meter interval, while SWE and density were calculated from observations at 25-50 m interval along the same snow course (i.e. usually 3-5 observations per snow course). We updated the dataset to add the mean HS at the same locations where the SWE and density were sampled. The measurements are now consistent as shown by the comparison between the density values and the ratio of the SWE to HS ratio (Fig. 2). The density computed by averaging the density values obtained in one snow course is not necessarily equal to the density computed as the ratio of the average SWE to HS for the same snow course. This non-linear effect is exacerbated in cases where the snowpack is high variable in space. The data including the HS from the snow tube version is provided at the following temporal link (https://www.dropbox.com/sh/zs9klzhl55ezdss/AAB-0wgKfHGheMmE05ovmFfpa?dl=0) and will be upgraded online with the revised version.

[Figure]

**Fig. 1.** Comparison of filtered incoming and outgoing shortwave radiation data at the sites

**Fig. 2.** Comparison of provided and computed snow density from snow core observations

---

## Referee Comment (RC2) · Anonymous Referee #2 · 18 Apr 2017

The paper presents in a very adequate way interesting meteorological, snow and re-mote sensing data in high elevated sites of Lebanon Mountains. The data may be very useful for many studies and helps to fill a big gap of knowledge on snow processes in key Mediterranean mountains. These areas deserve scientific attention as they are considered hot-spots in terms of likely climate change effects and because snow plays a major role in environmental and socioeconomic processes.

The paper is very well written and I think it suites very well in the on going special issue. I only have very minor comments that authors may consider for preparing a revised version of the manuscript.

- The manuscript states that treeline is at 1550 m a.s.l. I think this is due to very heavy human impact on vegetation and this is not the natural or climatic tree line, it can be

simply remarked. - In study are some information on temperature in the area or the location of annual and winter 0°C isotherm may be added.

- A little more discussion on how precipitation data can be affected by under-catch might be added. Perhaps with a simple statement relating the average wind speeds in the different sites with available literature. This could be added following the sentence finishing in line 188. - In line 286 you provide data from November to June and this is considered as "snow season". Afterwards, it is seen that snow normally lasts until March-April as the latest. Perhaps the provided data for the "snow period" should be reconsidered. - Line 306: Change m amsl by m a.s.l. - Line 338: Better to say the 25th and 75th percentiles. - Line 378: Snow height, not high - Line 424-426- What does mean the range of percentages? Is it the interannual variability (2011-2016) of the average SCA?

Looking forward to see the published manuscript.

---

## Author Comment (AC1) · 18 May 2017

Author response to comments on manuscript ESSD-2017-3 entitled "Snow observations in Mount-Lebanon (2011–2016)" by
Abbas Fayad et al.

We thank both referees for their comments which have substantially improved the clarity and accuracy of the manuscript. Our response to both referees are presented below:

- Referee comments are in bold and the author responses in normal font.
- Quotes from the text are italicized and the proposed revisions are underlined.
- Line numbers are referenced to the original manuscript.

**Anonymous Referee #1**

**RC1 General Comments**
**The authors present a meteorological and snow dataset collected over 2011-2016 in Mount-Lebanon. This includes 30-min meteorological observations at three sites, snow course measurements of snow depth, SWE, and density at 30 locations, and 500-m snow covered area and snow covered duration maps from MODIS. Given the lack of snow data in this region of the world, the paper has the potential to make a novel contribution. I could envision these sites being used in a snow model intercomparison study, which are often conducted across diverse sites. That being said, there are many major aspects that require the authors' attention before this should be considered for publication in ESSD.**

We thank the anonymous referee # 1 for his thoughtful review of our paper and we have addressed the comments below.

**MAJOR COMMENTS**

1) **In the provided dataset, there are periods when the outgoing shortwave data exceed the incoming shortwave data. This may occur during periods when snow is covering the up-pointing pyranometer but the down-pointing pyranometer is snow-free (e.g., see Lapo et al. 2015). While the text suggests that the incoming and reflected shortwave radiation measurements were screened by constraining albedo between 0 and 1, the provided data do not support this claim. The attached figure shows the problem. This needs further attention.**

Thank you for pointing this issue. We provided the raw incoming and outgoing solar radiation data by mistake, instead of the processed data as indicated in the manuscript. The radiation data were flagged using the following classes (0: night time with no shortwave observations; 1: data are correct, albedo test is OK (is between 0 and 1); 2: data do not pass the albedo test; 3: observation bias likely due to frost on the sensor; 4: field observations reported that the sensor was not well levelled; 5: no data). Extracting shortwave data with flag 1 only results in albedo

values that are within 0 and 1. We also used visual inspection to check for potential biases arising from snow frost especially when the outgoing shortwave exceeded the incoming SW during winter season. The corrected data is shown in Fig. A.1. and can be accessed following this temporal link (https://www.dropbox.com/sh/zs9klzhl55ezdss/AAB-0wgKfHGheMmE05ovmFfpa?dl=0). The data in the public repository will be upgraded with the revised version of the manuscript.

2) **The provided snow course data exhibit a fundamental inconsistency. The bulk snow density (with respect to water density) is the ratio of SWE by snow depth (as in Equation 5). However, when I compute this value from SWE and snow depth and compare to the density values in the dataset, they do not match (see attached figure). Please revisit the data and correct this issue. Because figures and analysis revolves around these data, it is essential to rectify these problems.**

The apparent inconsistencies in the snow height (HS), snow water equivalent and density are due to different sampling strategies during a snow course. We provided HS that is the average of the HS measurements collected at 5 meter interval, while SWE and density were calculated from observations at 25-50 m interval along the same snow course (i.e. usually 3-5 observations per snow course). We updated the dataset to add the mean HS at the same locations where the SWE and density were sampled. The measurements are now consistent as shown by the comparison between the density values and the ratio of the SWE to HS ratio (Fig. A.2). The density computed by averaging the density values obtained in one snow course is not necessarily equal to the density computed as the ratio of the average SWE to HS for the same snow course. This non-linear effect is exacerbated in cases where the snowpack is high variable in space. The data including the HS from the snow tube version is provided at the following temporal link (https://www.dropbox.com/sh/zs9klzhl55ezdss/AAB-0wgKfHGheMmE05ovmFfpa?dl=0) and will be upgraded online with the revised version of the manuscript.

3) **The sites do not have meteorological data that are gap-free and consequently are not immediately useful to modelers and others looking for sites to run/test models. The convention of many other snow data papers has been to provide data that are complete in time, using various in-filling techniques. The data presented here would be of greater value if the gaps were filled.**

The authors agree that it would be great to have gap-free meteorological data. Given the fact that this the first dataset in this region, we tried as much as possible to use proper techniques to flag and remove erroneous values. The exercise of gap-filling the data e.g. using linear interpolation or multiple linear regression (e.g., Kormos et al., 2017) remains challenging due to the limited number of stations and the high difference in elevation (elevation range between 1840 and 2834) between the three stations. We do not know yet who will download and use the data but we anticipate that the data can be used to validate, calibrate, or even downscale the output of an atmospheric model. In this case, it would not be recommended to use gap-filled values. Therefore, we think that the gap-filling should be let up to the data user. Actually, we have already dealt with this issue for a subsequent snow modelling study. In our case, we have

used MicroMet (Liston and Elder 2006) that employs a variety of procedures including autoregressive and spatial interpolation techniques to create distributed atmospheric fields while accounting for known temperature–elevation, precipitation–elevation, and wind–topography relationships. We can provide these data in addition to the quality checked data if you think that it would be appropriate for this paper.

**GENERAL COMMENTS**

1) **Are the temperature sensors naturally or mechanically ventilated? Please note this in section 3, perhaps in the paragraph on page 4, starting at line 25. To enhance the value of your dataset, you could consider preparing a corrected temperature dataset based on your reflected shortwave dataset based on Huwald et al. (2009).**

   The temperature sensors are naturally ventilated and installed inside radiation shield of the type "RAD10 10-Plate Shield" (https://s.campbellsci.com/documents/fr/manuals/cs215.pdf). The manuscript was updated, page 4 line 27: *"The temperature sensors are protected against solar radiation (Huwald et al., 2009) using radiation shield and are naturally ventilated"*.

2) **While the sites do provide valuable meteorological and snow data in a unique environment, I would advise the authors to note that the sites do not measure incoming longwave radiation, a key variable for the energy balance of warm, Mediterranean climates.**

   We thank the referee for this comment. The manuscript at page 4 lines 8-9 was updated to highlight that we are measuring only shortwave radiation. We also included at line 9 the importance of measuring longwave radiation in Mediterranean regions.

   *"Meteorological data, including snow depth, temperature, relative humidity, incoming and reflected shortwave solar radiation, wind speed and direction, and atmospheric pressure, are collected at the three sites using sensors mounted to towers (Fig. 3). Longwave radiation which are an important component of the energy balance in Mediterranean regions (Herrero and José Polo 2016) are not measured. However, incoming longwave radiation can be estimated from air temperature and humidity measurements at the stations (Brutsaert 2013) . Incoming solar radiation can also be used to better constrain the cloud fraction in the computation of the longwave radiation"*.

3) **Please include the elevation and coordinate (e.g. latitude/longitude) data for each of the 30 snow course sites.**

   The snow course metadata was updated to include information on the snow course coordinate (latitude/longitude) and elevation.

4) **Some of the figures are difficult to read because of low resolution or text fonts that are too small. These include Figures 4, 7, and 9.**

   We have made changes to Figures 4, 7, and 9 as suggested.

5) **In numerous places (e.g., P.7, L.21), the authors use the word "weighting" when they should actually be using the word "weighing" (no "t"). These words have different Meanings.**

The word was replaced accordingly (Page 7 lines 21, 32 and 33).

6) **The procedures for measuring snow depth, SWE, and bulk density with the federal sampler (P. 7, L.19-26) are fairly standard, so this section may be providing too much detail and can be greatly reduced.**

We agree that the methodology is standard especially in the USA. However, we feel the information provided may be helpful for non-US regions where snow measurements are often measured using non-standardized approaches.

**TECHNICAL CORRECTIONS**

We thank the referee for the elaborated technical comments that were updated accordingly in the text.

- P.1, L.15: "Precipitation data is" was replaced with "Precipitation data were".

- P.2, L.20: the phrase was updated to "lack of an operational snow observation network in Lebanon"

- P.2, L.24: Replaced "were made" with "are". The phrase now reads "groundwater recharge and streamflow are available for basin scale studies"

- P.2, L.32: The phrase was updated to define the acronym "AWS" and now reads "continuous meteorological and snow height observations collected at three automatic weather stations (AWS)"

- P.3, L.12: Replaced "covers" with "and covering".

**- P.4, L.15-16: The sentence is confusing. It can mean that each of the sites have temperature/ humidity sensors at three heights above ground, or that these are the heights at the three sites, respectively. I think it is the second meaning. Please rephrase for clarity.**

The phrase was updated: "Temperature and humidity sensor are installed at 2.4 m in MZA, 3.9 m in LAQ, and 4.2 m in CED."

- P.4, L.19: "is being observed" replaced with "has been observed".
- P.5, L.14: Replaced "was adjusted" with "were adjusted".
- P.5, L.17: Replaced "was removed" with "were removed".
- P.5, L.22: Replaced "is greater" with "was greater".
- P.5, L.27: Added event after precipitation to read "precipitation event".
- P.5, L.31-32: Replaced "is" with "was" to read "was equal", "was assumed", and "was preserved".

- P.7, L.2: Used a subscript for the zero on "$z_0$".
- P.7, L.16: the sentecne was updated to read "Field measurements of snow depth, snow density, and SWE were carried"
- P.7, L.19: Added "a" before "federal snow sampler".
- P.7, L.22: Replaced "recorder" with "recorded".
- P.8, L.21: Reference to Fig. 9 was removed.
- P.8, L.24-25: Replaced "starting the snow season 2014-2015" with "starting in snow season 2014-2015".

**P.8, L.28: Considering the measurement precision of the temperature sensors, I think it is appropriate to only reference to the nearest tenth of a degree here (and elsewhere). Please replace "6.93, 4.26, and -1.36" with "6.9, 4.3, and -1.4". The same comment holds for the wind speed averages (P.9, L.2-3).**

All values were rounded to the first decimal as suggested.

- P.9, L.1: Replaced "recorder" with "recorded".
- P.9, L.10: Replaced "Rain on snow event is" with "Rain on snow events are".
- P.10, L.7: Replaced the word "supportable" with "compacted" to read "snow is usually wetter and less compacted when compared to high elevation regions"

- P.10, L.9: Added "a" before "few days".
- P.10, L.10: Changed "felt" to "fell".

**- P.10, L.16-17: These are backwards. Fig. 8a shows HS vs. SWE while Fig. 8b shows HS vs. density. Please correct.**

The phrase was updated to read "Fig. 8. shows the general relationship between SWE vs HS (Fig. 8a) and density vs HS (Fig. 8b)."

- P.10, L.20: Added "an" before "approach".
- P.10, L.23: Added "a" before "nonlinear".
- P.10, L.24: Changed "account" to "accounts".
- P.10, L.30: Changed the subscript "i" in "hi".
- P.11, L.12: Changed "snow falls " with "snow storm events".
- P.12, L.13: Deleted "are" after "AWS data".
- P.12, L.14: Changed "potentials" to "potential".
- P.16, L.18: Replaced "show" with "shown".
- P.16, L.23: Replaced "till" with "until" to avoid slang.
- P.19, L.12: Started a new sentence after the parentheses, i.e.: "No correction for the accumulation: : :". To read "(a) Significant evaporation occurred during winter season 2014 (e.g., MZA: 12 – 29 December, 2013) and required manual correction (28 – 29 December). No correction for the accumulation of raw precipitation between 22 – and 23 December was made because the observed average humidity was below 15%."
- P.24, L.21: Replaced "Tabel" with "Table".

**TABLE AND FIGURE COMMENTS**
**- Figure 2: Please add "(a)" and "(b)" to the panels of this figure.**

We have made this change as suggested.

**- Figure 4: For clarity, please add either a legend to differentiate the different datasets or a note in the caption to identify which data pair with each vertical axis.**

We have updated Figure 4 to differentiate the different datasets as suggested.

**- Figure 5: Please add "(a)" and "(b)" to the panels of this figure.**

Was updated accordingly.

**- Figure 6: Please add "(a)","(b)", and "(c)" to the panels of this figure.**

Was updated accordingly.

**- Figure 7: Please add "(a)","(b)", and "(c)" to the panels of this figure. You may need to include "(d)", "(e)", and "(f)" as these are referenced in the text.**

The caption was updated to read *"Figure 7: Box and whisker plots for non–zero data: (a) snow depth, (b) SWE, and (c) snow density over snow season 2014–2015 (n = 311) and (d) snow depth, (e) SWE, and (f) and snow density over snow season 2015–2016 (n = 371) distributed by region where CED (elevation range, 1650–2900 m), LAQ (1300 – 1850 m) and MZA (1350 – 2350 m). The box brackets are 25% and 75% of the data (lower and upper boxes respectively). The whiskers are at minimum and maximum values."*

**- Figure 7: The caption says that (b) is the snow density, but the middle plots show SWE. Please correct either the caption or the ordering of the panels.**

Was updated accordingly.

**REFERENCES**
Huwald, H., C. W. Higgins, M.-O. Boldi, E. Bou-Zeid, M. Lehning, and M. B. Parlange (2009), Albedo effect on radiative errors in air temperature measurements, Water Resour. Res., 45(8), W08431, doi:10.1029/2008WR007600.
Lapo, K. E., L. M. Hinkelman, C. C. Landry, A. K. Massmann, and J. D. Lundquist (2015), A simple algorithm for identifying periods of snow accumulation on a radiometer, Water Resour. Res., 51(9), 7820–7828, doi:10.1002/2015WR017590.
Liston, G.E., Elder, K., 2006. A meteorological distribution system for high-resolution terrestrial modeling (MicroMet). J. Hydrometeorol. 7, 217–234.
Brutsaert, W. (2013). Evaporation into the atmosphere: theory, history and applications (Vol. 1). Springer Science & Business Media.

**Anonymous Referee #2**

**RC1 General Comments**

**The paper presents in a very adequate way interesting meteorological, snow and remote sensing data in high elevated sites of Lebanon Mountains. The data may be very useful for many studies and helps to fill a big gap of knowledge on snow processes in key Mediterranean mountains. These areas deserve scientific attention as they are considered hot-spots in terms of likely climate change effects and because snow plays a major role in environmental and socioeconomic processes. The paper is very well written and I think it suits very well in the on going special issue. I only have very minor comments that authors may consider for preparing a revised version of the manuscript.**

We thank the referee for a thorough review and we have addressed the comments below.

1) **The manuscript states that treeline is at 1550 m a.s.l. I think this is due to very heavy human impact on vegetation and this is not the natural or climatic treeline, it can be simply remarked.**

   We added page 3 line 18: *"The retreat of the natural tree line is due to the increased deforestation and urbanization. The natural tree line which can be still found in sparse small forested regions extends up to 2450 m in Abou Ali and 1900 m in Ibrahim and El Kelb Basins."*

2) **In study area some information on temperature in the area or the location of annual and winter 0C isotherm may be added.**

   To our knowledge no information on the winter 0C isotherm was published for this region.

3) **A little more discussion on how precipitation data can be affected by under-catch might be added. Perhaps with a simple statement relating the average wind speeds in the different sites with available literature. This could be added following the sentence finishing in line 188.**

   Page 6 Line 27 we added the following statement *"The median wind speed recorded during precipitation events over the winter seasons 2014-2016 ranged between 3.3 and 3.7 m s-1 for LAQ and 8.8 and 9 m s-1 for MZA with maximum recorded wind speed at 10 and 20.1 m s-1 for LAQ and MZA respectively."*

4) **In line 286 you provide data from November to June and this is considered as "snow season". Afterwards, it is seen that snow normally lasts until March-April as the latest. Perhaps the provided data for the "snow period" should be reconsidered.**

   The reason for considering the snow season between November to June is attributed to the fact that snowfall occurs between November till late March early April but snowmelt continues

through May (this can be seen in Figure 2 and 5). We frequently observed significant snow patches up to June during field visits.

5) Page 9 Line 10 (Line 306): Changed m amsl to m a.s.l.

6) Line 338: Better to say the 25th and 75th percentiles.

7) Page 10 line 2: Changed "representing the first 25% and last 75%" to "representing the the 25th and 75th percentiles".

8) **Line 378: Snow height, not high**

Was updated accordingly.

9) **Line 424-426- What does mean the range of percentages? Is it the interannual variability (2011-2016) of the average SCA?**

Yes, the range represents the interannual variability (2011-2016) and we updated the statement at page 11 Line 15 to read "winter months (December – March), for the years between 2011 and 2016,".

[Figure]

[Figure]

Figure 4: Example of daily precipitation and temperature observations at Laqlouq (1840 m a.s.l.) during snow season 2015-2016 (November–June).

[Figure]

**Figure 7:** Box and whisker plots for non–zero data: (a) snow depth, (b) SWE, and (c) snow density over snow season 2014–2015 (n = 311) and (d) snow depth, (e) SWE, and (f) snow density over snow season 2015–2016 (n = 371) distributed by region where CED (elevation range, 1650–2900 m), LAQ (1300 – 1850 m) and MZA (1350 – 2350 m). The box brackets are 25% and 75% of the data (lower and upper boxes respectively). The whiskers are at minimum and maximum values.

[Figure]

Figure 9: From left to right: time series of the snow cover area (SCA, percentage of the basin area), boxplot of mean monthly SCA, and SCD for (from top to bottom) Abou Ali, Ibrahim, and El Kelb River Basins.

[Figure]

Fig. A.1 Comparison of filtered incoming and outgoing shortwave radiation data at the sites

[Figure]

Fig. A.2. Comparison of provided and computed snow density from snow core observations